# Well Begun is Half Done: Generator-agnostic Knowledge Pre-Selection for Knowledge-Grounded Dialogue

**Lang Qin**[1,2]  **Yao Zhang**[*3]  **Hongru Liang**[4]  **Jun Wang**[5] and **Zhenglu Yang**[*1,2,6]

[1]TKLNDST, CS, Nankai University [2]Key Laboratory of DISSec, Ministry of Education, China
[3]School of Statistics and Data Science, LPMC, KLMDASR & LEBPS, Nankai University
[4]College of Computer Science, Sichuan University
[5]College of Mathematics and Statistics Science, Shandong Key Laboratory of
Language Resource Development and Application, Ludong University
[6]Education Field Integrated & Publishing Knowledge Mining & Service Key Laboratory,
National Press and Publication Administration, China
qinlang14@mail.nankai.edu.cn, yaozhang@nankai.edu.cn, lianghongru@scu.edu.cn,
junwang@mail.nankai.edu.cn, yangzl@nankai.edu.cn

## Abstract

Accurate knowledge selection is critical in knowledge-grounded dialogue systems. Towards a closer look at it, we offer a novel perspective to organize existing literature, i.e., knowledge selection coupled with, after, and before generation. We focus on the third under-explored category of study, which can not only select knowledge accurately in advance, but has the advantage to reduce the learning, adjustment, and interpretation burden of subsequent response generation models, especially LLMs. We propose GATE, a generator-agnostic knowledge selection method, to prepare knowledge for subsequent response generation models by selecting context-related knowledge among different knowledge structures and variable knowledge requirements. Experimental results demonstrate the superiority of GATE, and indicate that knowledge selection before generation is a lightweight yet effective way to facilitate LLMs (e.g., ChatGPT) to generate more informative responses.

## 1 Introduction

Knowledge-grounded dialogue systems generate informative responses by incorporating external knowledge, such as unstructured documents and structured knowledge graphs (Ghazvininejad et al., 2018; Lian et al., 2019). This generation process requires an agent to select context-related knowledge to support high user engagement. Taking Figure 1 (a) as an example, the knowledge "*Mark Boal wrote Zero Dark Thirty*" contributes to a high-quality response compared with the knowledge "*Zero Dark Thirty has genre War film*", when the user focuses on "*who wrote Zero Dark Thirty*".

Many efforts have been devoted to selecting context-relevant knowledge, and we classify them into three categories based on the occasion when knowledge selection is performed. The first category is co-selection (Zhao et al., 2020; Tuan et al., 2022; Bai et al., 2023) wherein knowledge selection and response generation are executed in a coupled manner (c.f. Figure 1 (b)-①). Although this category of approach is efficient and has been extensively researched, it is costly to learn and is difficult to interpret and adjust when errors arise in the generated responses. The second category is post-selection (Dziri et al., 2021; Xue et al., 2022), namely, the knowledge is selected after the generation and is used to correct the knowledge error in the generated response (c.f. Figure 1 (b)-②). This category is skilled in adjusting local errors yet has minimal impact on enhancing the informativeness of the response.

Surprisingly, the third category, pre-selection, has been under-explored by previous research. This category performs knowledge selection as an independent model before response generation (c.f. Figure 1 (b)-③). In this paper, we pay attention to this study, not only because of the overlook by current work but also because selecting knowledge accurately in advance can provide the potential to reduce the learning, adjustment, and interpretation burden of subsequent response generation models, especially for large language models (LLMs). To select knowledge for preparation, in addition to the accuracy required for knowledge selection, there are two key issues that need to be addressed:

- *Different knowledge structure*. An ideal knowledge selector should be able to tackle different knowledge structures, such as unstructured knowledge represented by text and structured knowledge represented by knowledge graphs.

- *Variable knowledge requirement*. Typically, the

---
[*]Corresponding author.

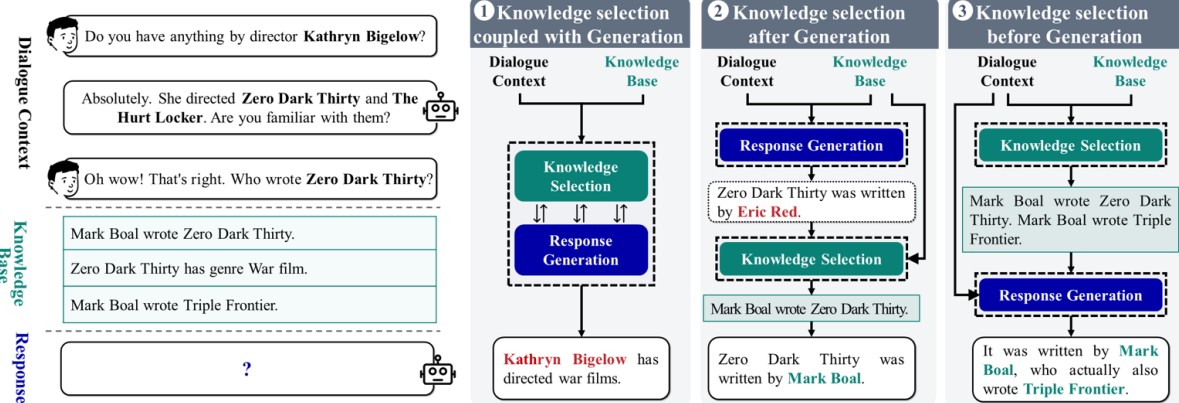

**(a) Example of knowledge-grounded dialogue**      **(b) Three categories of knowledge selection**

Figure 1: a) An example of knowledge-grounded dialogue. At each turn, the dialogue agent generates a response by selecting knowledge from the knowledge base. b) Three categories of knowledge selection used in the knowledge-grounded dialogue system: ① knowledge selection coupled with generation (co-selection), ② knowledge selection after generation (post-selection), and ③ knowledge selection before generation (pre-selection). Notice that the dashed rectangular box refers to an independent model.

number of knowledge required to generate an informative response is neither one nor a fixed number but a variable number. For example, when the dialogue is about an award-winning film rather than a little-known one, there is much more relevant knowledge involved. An ideal knowledge selector should be able to dynamically adapt the number of selected knowledge.

To resolve the above issues, we propose GATE, a Generator-AgnosTic knowledgE selection method to prepare knowledge for subsequent various response generation models, e.g., BART (Lewis et al., 2020) and T5 (Raffel et al., 2020). GATE has the ability to confront both unstructured and structured knowledge and to adapt the number of desired knowledge according to different dialogue contexts. Specifically, GATE consists of: knowledge structure unifying, knowledge scoring, and knowledge pool size adapting module. Further, we employ a reinforcement learning (RL) framework to train GATE, optimizing the reward of selecting appropriate knowledge in both quality and quantity. The experimental results on two datasets demonstrate the superiority of GATE on knowledge selection, and that GATE can facilitate the response generation model (including ChatGPT) to generate more informative responses. We believe that GATE provides a lightweight and efficient solution, which is a potentially viable way for reducing the learning, adjustment, and interpretation burden of LLMs. In summary, the main contributions of this work are as follows.

• We introduce a novel perspective to organize the

literature of knowledge selection in knowledge-grounded dialogue, i.e., knowledge selection coupled with, after, and before generation. Besides, we point out that the third category of study, though under-explored, has advantages to reduce the learning, adjustment, and interpretation burden of subsequent response generation models.

• We propose GATE, a generator-agnostic knowledge selection method, to prepare knowledge for subsequent response generators by selecting context-related knowledge among different knowledge structures and variable knowledge requirements.

• We conduct experiments to demonstrate the superiority of GATE, and find that knowledge pre-selection is a lightweight and effective way to facilitate ChatGPT to generate more informative responses.

## 2 Related Work

Knowledge selection is a crucial step in the knowledge-grounded dialogue system. Our work provides a new perspective to review the literature of knowledge-grounded dialogue——based on different time points of knowledge selection in the response generation process (c.f., Figure 1(b)).

**Knowledge selection coupled with Generation** This knowledge selection category denotes that the knowledge selection and response generation processes are modeled to be executed concurrently in a single model, as shown in Figure 1 (b)-①. For unstructured knowledge, the Co-Selection process is an interactive matching process between the

dialogue and the documents (Meng et al., 2020, 2021). Dinan et al. (2019) utilizes dot product attention to select the most relevant knowledge. Bai et al. (2023) improves selection by enhancing knowledge's dense representation. For structured knowledge, the selection process can be viewed as a multi-hop reasoning procedure on a graph, which is subsequently followed by a two-stage architecture for response generation (Liu et al., 2019; Zhou et al., 2021; Tuan et al., 2022). However, the construction or training of the above methods is tied to the generation model. In contrast, GATE is "plug-and-play" and can enhance response generation for various generation models.

**Knowledge selection after Generation** This knowledge selection category denotes that knowledge selection is executed as an independent model after response generation, as shown in Figure 1 (b)-②. Post-selection is dedicated to correcting potential knowledge errors in the response. Dziri et al. (2021) address hallucinations in responses by replacing them with correct knowledge obtained from the knowledge graph. Xue et al. (2022) employ text infilling to incorporate retrieved knowledge into incomplete responses. However, these methods would diminish the fluency and naturalness of responses, whereas GATE does not compromise generation models.

**Knowledge selection before Generation** This knowledge selection category denotes that knowledge selection is executed as an independent model before response generation, as shown in Figure 1 (b)-③. Pre-selection is dedicated to improving the accuracy of knowledge selection and further enhancing the quality of response generation. A few works have actually employed Pre-Selection without emphasizing or providing a formal definition. Jung et al. (2020) implement graph-based reasoning through attention flows to select knowledge (i.e., paths in the graph). Eric et al. (2021) collected an augmented dataset and proposed a ranking-generation pipeline to evaluate it. Li et al. (2022) constructs semantic graphs of textual knowledge and performs selection based on node similarity. Yang et al. (2022) proposed a topic-shift aware knowledge selector to utilize the role-initiative information to help select knowledge. However, the above methods are designed to address specific knowledge types, while GATE can select knowledge across different knowledge bases.

In summary, existing methods are constrained to specific knowledge types or generation models, significantly limiting the generalization ability, and rely on a fixed-size knowledge pool, which could undermine the performance of response (Shuster et al., 2021). In contrast, GATE operates in a Pre-Selection category that can handle diverse knowledge types and adapt the knowledge pool size to enhance various generation models.

## 3 Methodology

### 3.1 Overview

We first formulate the knowledge-grounded dialogue as follows: at $t$-turn conversation, given the dialogue history $\mathcal{X}_t$ and currently accessible knowledge base $\mathcal{K}$, the system generates response $\mathcal{Y}_t$ based on the dialogue-relevant knowledge set $\mathcal{K}_t \subseteq \mathcal{K}$. Then, the knowledge pre-selection task that we focus on refers to how to select a more useful knowledge set $\mathcal{K}_t^*$ from the knowledge set $\mathcal{K}_t$ before the response generation process. "Useful" here means that $\mathcal{K}_t^*$ can help the agent generate high-quality responses.

To select a more useful knowledge set $\mathcal{K}_t^*$, GATE performs three efforts:

- Unifying knowledge of diverse structure types (unstructured and structured), to improve the generalization and flexibility of GATE.

- Scoring knowledge to help select the knowledge that is more relevant to the desired response.

- Adapting knowledge pool size to provide appropriately sized knowledge sets for subsequent response generation models.

We will introduce the details of GATE in Section 3.2, the reinforcement learning framework for GATE in Section 3.3, and the optimization and training details of GATE in Section 3.4.

### 3.2 GATE

**Knowledge Structure Unifying** GATE first unifies knowledge of diverse structure types to improve its generalization and flexibility. In general, there are two types of knowledge structures used by the knowledge-grounded dialogue system: unstructured and structured knowledge. Unstructured knowledge generally exists in the form of documents, and structured knowledge typically uses knowledge graphs. Graph structures are lengthy in modeling the association information between

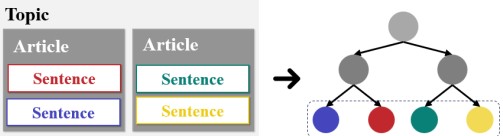

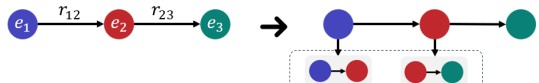

Figure 2: Knowledge Structure Unifying

knowledge and can help to select useful knowledge more accurately. Moreover, graph structures can enhance the interpretability of the knowledge selection process. Therefore, GATE uniformly transforms all the diverse types of knowledge into a graph structure, which has two types of nodes, i.e., process node and knowledge node.

For document-based unstructured knowledge, it naturally has a $topic \gg article\ title \gg sentence$ hierarchy. GATE presents this hierarchy as a graph, where topics and article titles are used as process nodes in the graph, sentences are used as knowledge nodes, and nodes are connected to nodes by edges if there is a containment relation, as shown in Figure 2 (a). For the structured knowledge graph, GATE keeps its original structure unchanged, and all the original nodes are used as process nodes. GATE will additionally add knowledge nodes [1]. For the triple $(e_i, r_{ij}, e_j)$ formed by two entity nodes ($e_i$ and $e_j$) and the relation edge ($r_{ij}$) between them, GATE merges them into a single knowledge node and connects them to the head entity ($e_i$) in this triple, as shown in Figure 2 (b).

**Knowledge Scoring** After unifying the knowledge structure, GATE next scores each process node in the graph to help select the knowledge that is more relevant to the desired response.

Firstly, a subgraph $Adj_t$ is obtained based on the valid state transition target of the Agent. As shown in Figure 3, the encoding of the Agent's state is concatenated with the encoding of each process node in the subgraph. We utilize a Graph Attention Network (GAT) (Brody et al., 2022) to score each node and then sample to obtain the target for state

---

[1] It has been demonstrated that using fact triples can help dialogue systems generate high-quality responses better than separate entities and relations. (Dziri et al., 2021; Sarkar et al., 2022).

transition:

$$\text{score}_n = \text{GAT}\left([\boldsymbol{S}_t; \boldsymbol{e}_n] \mid \text{for } n \text{ in } Adj_t\right). \quad (1)$$

The knowledge nodes of sorted process nodes form $\mathcal{K}_t$. Considering the guiding role of process node scores in knowledge selection, we calculate node attention weights based on the score distribution using MLP. These weights are utilized in dot-product calculations with the Agent state and knowledge encoding to determine the score of each knowledge in $\mathcal{K}_t$.

**Knowledge Pool Size Adapting** GATE determines the knowledge pool's appropriate size by analyzing the node score distribution variance. The top-ranked knowledge is selected to constitute $\mathcal{K}_t^*$, where $\mathcal{M}(\cdot)$ maps the input to the interval [0,1]:

$$|\mathcal{K}_t^*| = |\mathcal{K}_t| * \mathcal{M}\left(\frac{1}{1 - \text{Var}(\text{score}_n)}\right). \quad (2)$$

### 3.3 RL Framework for GATE

Graph-based reasoning improves the interpretability of knowledge selection. Due to the extensive use and success of reinforcement learning in this field, we formulate the knowledge selection process as a Markov Decision Process and employ reinforcement learning for graph-based reasoning.
**State** We employ SentenceBert (Reimers and Gurevych, 2019) to perform static encoding of the nodes and knowledge within the graph structure outlined in Section 3.2. We encode each piece of knowledge attached to a node as $\{e_{k_i}\}_{i=1}^{|e_{k_i}|}$, and use the mean-pooling of these encodings as the node's encoding: $e_n = \text{MeanPool}\left(\{e_{k_i}\}_{i=1}^{|e_{k_i}|}\right)$. We leverage KeyBert (Grootendorst, 2020) to extract keywords $\mathcal{W}_t$ from the conversation history $\mathcal{X}_{1:t-1}$ and user statement $\mathcal{X}_t$. Then, we use a modified multi-head hierarchical modality attention mechanism (Moon et al., 2018) to process all input information: $\overline{\boldsymbol{x}} = \text{Attention}(\mathcal{X}_{1:t-1}, \mathcal{X}_t, \mathcal{W}_t)$.

After initializing and encoding the graph, we employ GAT to update the encoding of the entire graph at the beginning of the Agent's traversal: $\{e_{n_i}\}_{i=1}^{|e_{n_i}|} = \text{GAT}\left(\{e_{n_i}\}_{i=1}^{|e_{n_i}|}\right)$.

As the Agent traverses the graph, we update its state considering the node $e_{n_t}$ that the Agent is located at time t:

$$\boldsymbol{S}_t = \text{Attention}\left(\overline{\boldsymbol{x}}; \boldsymbol{S}_{t-1}; \boldsymbol{e}_{n_t}\right). \quad (3)$$

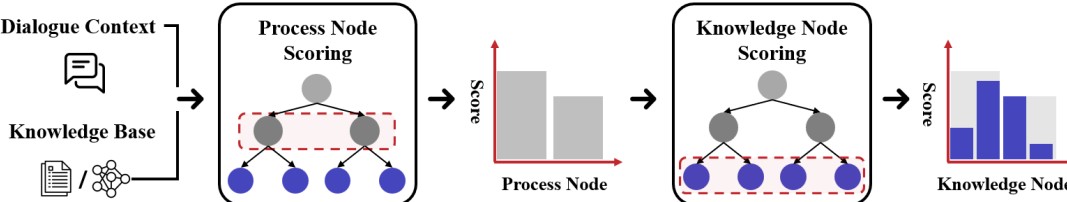

Figure 3: Knowledge scoring in GATE

**Action** The Agent's traversal on the graph is a Markov Decision Process with a maximum step count of $\mathcal{T}$. The Agent's action encompasses moving to the next node and constructing the corresponding knowledge pool. The action space is the set of one-hop neighbors of the current node.

**Reward** To accomplish our objective of enhancing the quality and quantity of knowledge selection, we optimize the model in three ways:

- Enhancing the Agent's ability to traverse the graph and reach the correct process node, we have $R_{Node}$ as positive if the Agent halts at the correct process node and negative otherwise.

- Improving the ability to select appropriate knowledge from the knowledge pool, we design $\mathcal{R}_{Gold}$:

$$\mathcal{R}_{Gold} = \text{Max}\left(1 - \alpha * \text{r}\left(k_{gold}\right), -1\right), \quad (4)$$

where r refers to the rank of ground-truth knowledge $k_{gold}$ in $\mathcal{K}_t^*$, and $\mathcal{R}_{Gold}$ is negative if not select $k_{gold}$.

- Adaptively determining an appropriate knowledge pool size. If the model selects the correct knowledge with a high probability, excessive knowledge can be deemed redundant. Otherwise, expanding the knowledge pool is necessary to enhance knowledge effectiveness. Hence, we design $\mathcal{R}_{Pool}$ as $\mathcal{R}_{Gold}/|\mathcal{K}_t^*|$.

The complete reward function is as follows:

$$\mathcal{R}_{s_t} = \mathcal{R}_{Node} + \mathcal{R}_{Gold} + \mathcal{R}_{Pool}. \quad (5)$$

**Policy Network** We design a policy network $\pi_\theta\left(a_t \mid s_t\right) = P\left(a_t \mid s_t; \theta\right)$ to obtain the probability distribution of actions, where $\theta$ represents the model parameters utilized in the Knowledge Scoring process and RL Framework.

### 3.4 Optimization and Training

The optimization objective of our policy network is to maximize the expected cumulative reward:

$$J(\theta) = \mathbb{E}_{a \sim \pi(a|s;\theta)}\left(\sum_t R_{s_t}\right). \quad (6)$$

We use the following stochastic gradient based on the REINFORCE algorithm (Williams, 1992) to optimize $\theta$:

$$\nabla_\theta J(\theta) \approx \nabla_\theta \sum_t R_{s_t} \log \pi_\theta\left(a_t \mid s_t\right). \quad (7)$$

In order to effectively utilize the available supervised information inherent in the graph, we integrate node loss $\mathcal{L}_{Node}$ and knowledge loss $\mathcal{L}_{Knowledge}$ using standard cross-entropy. We derive $\mathcal{L}_{Walk}$ by taking the negative value of $J(\theta)$. Then, the complete loss function is as follows:

$$\mathcal{L} = \mathcal{L}_{Walk} + \mathcal{L}_{Node} + \mathcal{L}_{Knowledge}. \quad (8)$$

The introduced model uniquely incorporates the aforementioned selection methods while maintaining its independence from any specific generator.

## 4 Experiments

We conduct experiments in knowledge selection and response generation to investigate the effectiveness of knowledge Pre-Selection in GATE. We also analyze the auxiliary capability of knowledge Pre-Selection on the generation model and the advantages of GATE adaptive knowledge pool size.

### 4.1 Datasets

We conduct experiments on Wizard of Wikipedia (WoW) (Dinan et al., 2019) and OpenDialKG (Moon et al., 2019) datasets for unstructured and structured knowledge bases. The statistics of the two datasets are presented in Appendix A.

In WoW, two participants take roles as a wizard and an apprentice. The wizard selects proper knowledge (sentence) from Wikipedia for the response. WoW split test set into Seen and Unseen based on topics. OpenDialKG is a parallel corpus comprising open-domain dialogues and a knowledge graph. The reasoning path of each turn is annotated, enabling participants to utilize graph information during the conversation.

Due to the absence of an official split in OpenDialKG, we follow WoW by dividing the dataset into Seen/Unseen categories based on topics.

| Method | PLM | Test Seen | | | | Test Unseen | | | |
|---|---|---|---|---|---|---|---|---|---|
| | | Rouge | Meteor | BLEU | F1 | Rouge | Meteor | BLEU | F1 |
| **WoW** | | | | | | | | | |
| Random | BART | 16.06 | 15.21 | 1.43 | 19.16 | 16.17 | 15.33 | 1.44 | 19.15 |
| Semantic | BART | 18.26 | 17.35 | 2.46 | 21.09 | 18.64 | 17.76 | 2.55 | 21.35 |
| TMNet | – | 16.34 | 13.87 | 0.85 | 17.49 | 14.98 | 12.25 | 0.51 | 13.20 |
| SKT++ | – | 18.68 | 17.33 | 2.43 | 19.83 | 17.09 | 14.75 | 1.68 | 17.46 |
| MIKE | – | 19.13 | 18.07 | 2.75 | 19.70 | 17.25 | 15.64 | 1.91 | 17.12 |
| KnowledGPT | GPT2 | 20.85 | 21.18 | 3.62 | 22.03 | 19.60 | 19.53 | 3.09 | 20.48 |
| KINET | BART | 21.47 | 21.63 | 3.84 | 22.45 | 20.54 | 20.48 | 3.47 | 21.45 |
| GATE | T5 | 20.17 ±0.19 | 20.36 ±0.19 | 3.75 ±0.14 | 22.34 ±0.20 | 19.05 ±0.13 | 18.98 ±0.13 | 3.15 ±0.08 | 21.28 ±0.12 |
| GATE | BART | **24.15** ±0.04 | **23.14** ±0.02 | **5.81** ±0.04 | **26.23** ±0.04 | **23.59** ±0.09 | **22.69** ±0.03 | **5.41** ±0.13 | **25.67** ±0.06 |
| **OpenDialKG** | | | | | | | | | |
| Random | BART | 24.52 | 24.31 | 4.97 | 28.45 | 24.29 | 24.02 | 4.60 | 28.43 |
| Semantic | BART | 25.84 | 25.64 | 5.67 | 29.59 | 25.55 | 25.30 | 5.29 | 29.58 |
| DialKG* | BART | 25.91 | 25.86 | 5.97 | 29.71 | 25.32 | 25.01 | 5.19 | 29.43 |
| AttnIO-AS* | BART | 27.04 | 26.86 | 6.37 | 30.94 | 26.65 | 26.44 | 5.60 | 30.65 |
| DiffKG | T5 | 18.27 | 16.87 | 1.23 | 21.20 | 17.57 | 16.06 | 0.94 | 20.58 |
| NPH | GPT2 | 27.53 | 27.56 | 6.71 | 31.34 | 26.97 | 27.01 | 5.76 | 30.51 |
| GATE | T5 | 23.88 ±0.04 | 23.09 ±0.07 | 4.26 ±0.01 | 26.36 ±0.04 | 23.58 ±0.04 | 22.63 ±0.02 | 3.70 ±0.05 | 26.17 ±0.04 |
| GATE | BART | **28.57** ±0.01 | **28.46** ±0.03 | **7.20** ±0.05 | **32.21** ±0.05 | **27.93** ±0.08 | **27.78** ±0.08 | **6.41** ±0.05 | **31.57** ±0.07 |

Table 1: The evaluation results of response generation on WoW and OpenDialKG datasets. The baseline model results for WoW are reported from Bai et al. (2023). "*" denotes our re-implementation.

## 4.2 Baselines

We choose the following four types of baselines:
1) Trivial baselines

**Random**: it randomly selects knowledge from the candidate pool;

**Semantic**: it selects knowledge based on semantic similarity between dialogue and knowledge.

2) Methods using Co-Selection

**TMNet** (Dinan et al., 2019): it combines memory network architecture and Transformer to encode dialogue and knowledge for response generation;

**SKT++** (Chen et al., 2020): it utilizes a Posterior Information Prediction Module and a Knowledge Distillation-Based Training Strategy;

**MIKE** (Meng et al., 2021): it introduces an initiative discriminator for knowledge selection;

**KnowledGPT** (Zhao et al., 2020): it utilizes pre-trained language models and optimizes via unsupervised learning;

**KINET** (Bai et al., 2023): it introduces a negative-enhanced knowledge approximator and a curriculum knowledge sampler;

**DiffKG** (Tuan et al., 2022): it employs Transformer to generate relation sequences on KG and generates responses based on retrieved entities.

3) Methods using Post-Selection

**NPH** (Dziri et al., 2021): it retrieves correct entities by crafting a query signal propagated over a graph to refine hallucination in response.

4) Methods using Pre-Selection

**DialKG** (Moon et al., 2019): it models the symbolic transitions as structured traversal on KG and predicts entities with a graph path decoder;

**AttnIO** (Jung et al., 2020): it flexibly adjusts the nodes and edges of focus based on dialogue context via attention flow.

Among the aforementioned baselines, TMNet, SKT++, MIKE, KnowledGPT, and KINET are utilized for unstructured knowledge, whereas DialKG, AttnIO, DiffKG, and NPH are utilized for structured knowledge.

## 4.3 Metrics

GATE strives to improve the accuracy of knowledge selection and further enhance the quality of response generation. Following previous works (Jung et al., 2020; Meng et al., 2021; Bai et al., 2023), we employ 1) ROUGE (Lin, 2004), BLEU (Papineni et al., 2002), Meteor (Denkowski and Lavie, 2014), and unigram overlap (F1) (Dinan et al., 2019) to evaluate the quality of the generated responses; and

| Method | PLM | Test Seen | Test Unseen |
|---|---|---|---|
| **WoW** | | | |
| Random | BART | 1.52 | 1.47 |
| Semantic | BART | 6.57 | 6.87 |
| TMNet | Transformer | 22.50 | 12.20 |
| TMNet | BERT | 23.86 | 16.33 |
| SKT++ | — | 27.62 | 20.20 |
| MIKE | — | 28.41 | 21.47 |
| KnowledGPT | GPT2 | 28.00 | 25.40 |
| KINET | BERT | 29.38 | 27.05 |
| KINET | BART | 28.90 | 27.14 |
| GATE | T5 | **31.81** | **27.60** |
| | BART | ±0.28 | ±0.41 |
| **OpenDialKG** | | | |
| Random | BART | 0.93 | 0.66 |
| Semantic | BART | 13.13 | 13.44 |
| DialKG* | BART | 19.92 | 17.22 |
| AttnIO* | BART | 24.96 | 22.67 |
| DiffKG | T5 | 29.31 | 23.65 |
| GATE | T5 | **30.21** | **27.72** |
| | BART | ±0.15 | ±0.28 |

Table 2: The evaluation results (R@1) of knowledge selection on the WoW and OpenDialKG datasets. "*" denotes our re-implementation. TMNet results are reported from Meng et al. (2021).

| Method | Engaging | | Informative | |
|---|---|---|---|---|
| | WoW | OpenDialKG | WoW | OpenDialKG |
| **+5 Pieces of Knowledge** | | | | |
| ChatGPT | **44.0%** | 46.0% | 33.0% | 31.5% |
| + GATE | 42.0% | **47.0%** | **44.5%** | **42.0%** |
| **+10 Pieces of Knowledge** | | | | |
| ChatGPT | **46.5%** | 37.0% | 35.5% | 29.5% |
| + GATE | 42.5% | **55.0%** | **48.0%** | **49.5%** |

Table 3: The evaluation results of ChatGPT on 200 response pairs. The percentages represent the proportion of responses that outperform one another.

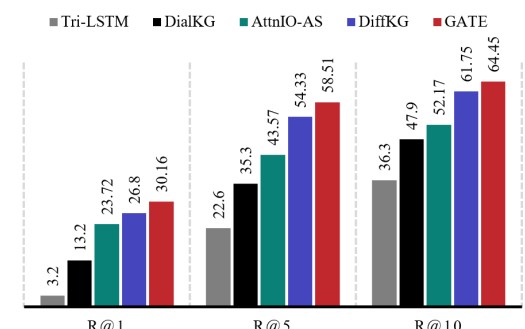

Figure 4: The evaluation results on the OpenDialKG origin data. The baseline model results are reported from Tuan et al. (2022).

2) Recall@k, which calculates the percentage of ground-truth knowledge included in the top-k selections, to evaluate the knowledge selection accuracy.

## 4.4 Implementation Details

We employ AdamW optimizer (Loshchilov and Hutter, 2017) with weight decay 0.12 and OneCycle policy (Smith and Topin, 2019). The activation function is LeakyReLU with a negative slope of 0.21. GATE is trained on a single RTX A5000. We use the same pre-trained checkpoints as the state-of-the-art approaches to ensure fairness. Specifically, we employ BART[2] and T5[3]. The code is available at https://github.com/qinlang14/GATE.

## 4.5 Results and Observations

**Overall Performance** Table 1 and Table 2 show that GATE markedly outperforms all baselines in terms of accuracy of knowledge selection and quality of the generated responses. To ensure the reliability of the results, we conduct additional experiments using the original data from OpenDialKG as Figure 4. We make the following observations:

- GATE outperforms previous SOTA methods in

---

[2] https://huggingface.co/facebook/bart-base
[3] https://huggingface.co/allenai/unifiedqa-t5-base

knowledge selection as measured by R@1. Our reinforcement learning-based graph reasoning method for unified knowledge types effectively selects relevant knowledge.

- GATE outperforms models utilizing the same generation module (e.g., Bart, T5) across multiple metrics. The knowledge selected by GATE significantly enhances generation modules.

- Specifically, in Table 1, the NPH method utilizing Post-Selection achieves considerable results through entity replacement but still falls short compared to using GATE's Pre-selection. Table 2 demonstrates that Co-Selection methods such as TMNet and KINET are constrained by the pre-trained models, whereas GATE can be generator-agnostic that selects relevant knowledge without being reliant on the generation model.

**Performance w.r.t. ChatGPT+GATE** To evaluate the auxiliary capability of our model, we input the knowledge selected by GATE into ChatGPT as supplementary information. We provide conversation history and instruct ChatGPT to generate responses using the selected knowledge or relying solely on its internal knowledge. Considering the cost of human evaluation, we follow recent works (Wang et al., 2023; Luo et al., 2023) to assess responses by ChatGPT. Detailed prompts are in Appendix B. The

| | |
|---|---|
| Dialogue Context | (User) I heard about a movie called The Fault in Our Stars. Who stars in it? (Agent) It's a good one! That was written by John Green and stars Shailene Woodley. (User) Oh, I didn't know that. What else has John Green Wrote? |
| Ground-truth Response | He also wrote Paper Towns. |
| Gold Knowledge | Paper Towns is written by John Green. |
| DiffKG | **K**: (A Confederacy of Dunces; written_by; Walker Percy) **R**: Sure! A Confederacy of Dunces is written by John Green. Do you like his work? |
| NPH | **K**: (Meg Crosbie; starred_actors; Paper Towns) **R**: Paper Towns is another one he wrote. It was released in Meg Crosbie. |
| GATE | **K**: (Paper Towns; written_by; John Green), (Paper Towns; release_year; 2008), ... **R**: Paper Towns is another one he wrote. It was released in 2008. |

Table 4: Case Study on OpenDialKG. "K"denotes selected knowledge and "R"denotes the generated response.

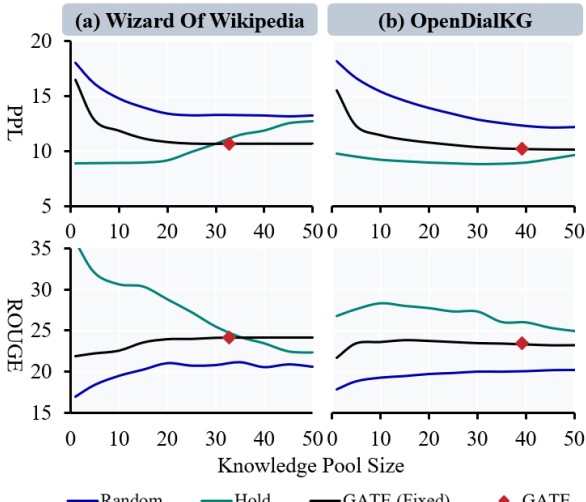

Figure 5: GATE adapting knowledge pool size at the sample level. "GATE(Fixed)" employs a fixed-size knowledge pool for comparison, "Random" randomly selects knowledge as a lower bound, and "Hold" ensures the correct knowledge is in the pool as an upper bound.

| Method | WoW | | OpenDialKG | |
|---|---|---|---|---|
| | Test Seen | Test UnSeen | Test Seen | Test UnSeen |
| **GATE** | **31.81** | **27.60** | **30.21** | **27.72** |
| w/o $\mathcal{L}_{Walk}$ | 30.42 | 26.19 | 29.26 | 26.90 |
| w/o $\mathcal{L}_{Node}$ | 31.23 | 27.09 | 26.68 | 24.53 |
| w/o $\mathcal{L}_{Knowledge}$ | 2.366 | 2.273 | 0.765 | 0.454 |
| w/ $\mathcal{L}_{Walk}$ | 0.894 | 0.491 | 0.573 | 0.717 |
| w/ $\mathcal{L}_{Node}$ | 2.471 | 2.247 | 0.526 | 0.669 |
| w/ $\mathcal{L}_{Knowledge}$ | 31.62 | 26.42 | 27.88 | 24.77 |

Table 5: The ablation results (R@1) for loss function.

results in Table 3 demonstrate that, although similar to the findings in Shuster et al. (2021) that utilizing redundant knowledge may impair engagement, the knowledge selected by GATE significantly enhances the information quality of responses. This improvement persists even when employing ChatGPT, already known for its powerful performance and vast implicit knowledge. Moreover, the inconsistency in engaging may stem from different knowledge quantities: the WoW dataset with complete sentences and overly detailed knowledge, causing rigid responses, and the OpenDialKG dataset with concise triplets, causing engaging responses.

**Performance w.r.t. Adaptive Pool Size** Figure 5 shows the response performance under different knowledge pool sizes and illustrates that GATE determines the appropriate size for dialogues. The range of knowledge pool sizes is up to 50 due to the limitation of input tokens that the generation model can accept. In WoW, sentences as knowl-

edge have higher informativeness, resulting in a relatively smaller number of required knowledge pieces. In OpenDialKG, triplets have less information, requiring a larger knowledge pool to obtain satisfactory results. "GATE(Fixed)" demonstrates that a certain amount of knowledge is necessary for better responses. "GATE" demonstrates that our method can select an appropriate number of knowledge pieces, resulting in high-quality responses. GATE offers improved flexibility and effectiveness compared to previous works that solely utilize a fixed and inadequate selection of top-k knowledge.

## 4.6 Ablation Study

As outlined in Section 3.4, the loss function comprises three components: walk, node, and knowledge. Table 5 presents the respective influence of each component on knowledge selection. Our model attains optimal performance by leveraging the synergistic effect of three components within the loss function. In particular, $\mathcal{L}_{Knowledge}$ serves as the universal optimization objective for this task, acting as the cornerstone for knowledge selection accuracy. $\mathcal{L}_{Walk}$ corresponds to the loss in the reinforcement learning of our model, while $\mathcal{L}_{Node}$ represents the loss in node selection after unifying the knowledge types. The combined influence of these two specialized components empower our model to achieve significantly higher accuracy than using $\mathcal{L}_{Knowledge}$ alone.

### 4.7 Case Study

As shown in Table 4, we conduct a case study on a sample from OpenDialKG. The dialogue topic shifted from movies to books, requiring the agent to provide recommendations of works by a specified author. DiffKG provides an appealing response but incorrectly "concatenates" the triples in its Co-Selection process, as the true author of "Confederacy of Dunces" is John Kennedy Toole. NPH utilizes Post-Selection based on the response from GATE, resulting in the disruption of sentence semantics, which is an inherent drawback of entity substitution methods. In contrast, GATE acquires relevant knowledge through Pre-Selection and produces an accurate and fluent response.

## 5 Conclusion

This paper offers a novel perspective to organize the literature on knowledge selection in knowledge-grounded dialogue systems, i.e., knowledge selection coupled with, after, and before generation. This paper focuses on the third category and proposes GATE, a generator-agnostic knowledge selection method, which prepares knowledge for subsequent response generation models by selecting context-related knowledge among different knowledge structures and variable knowledge requirements. The experimental results demonstrate the superiority of GATE on knowledge selection, and that GATE can facilitate the response generation model (including ChatGPT) to generate more informative responses.

## Limitation

Despite we have conducted experiments demonstrating the remarkable ability of GATE in improving the performance of ChatGPT and producing more informative responses, there is still ample scope for further exploration regarding the contribution of our model to LLMs. For example, We can try to combine GATE with more advanced prompts techniques developed recently (Wei et al., 2023; Tan et al., 2023; Lu et al., 2023) to facilitate more LLMs. We believe this will be an effective way to amplify the capability of GATE.

## Acknowledgements

This work was supported in part by the National Natural Science Foundation of China (No.62206191, No.62306156 and No.62106091); in part by the China Postdoctoral Science Foundation (No.2021TQ0222 and No.2021M700094); in part by the Natural Science Foundation of Sichuan (No.2023NSFSC0473); in part by the Shandong Provincial Natural Science Foundation (No.ZR2021MF054); and in part by the Fundamental Research Funds for the Central Universities, Sichuan University (No.2023SCU12089) and Nankai University (No.63231183).

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

## A    Dataset Statistics

As shown in Table 6, WoW has 22,311 conversations and a test set split into Seen and Unseen based on topics. Consistent with previous research, we employ the knowledge retrieved for the last two turns as the knowledge pool. OpenDialKG has 15,673 conversations and a knowledge graph that contains 100K entities and 1.1M facts. The reasoning path of each turn is annotated, enabling participants to utilize graph information during the conversation. We maintain the train/valid/test sets in a 70%/15%/15% ratio, consistent with previous works.

## B    Prompts for ChatGPT

Tables 7 and Tables 8 contain the prompts for generation and evaluation to derive the results presented in Table 3.

## C    Supplementary Ablation Study

Table 9 presents the extended results of Table 5, measuring the performance on R@1/5/10. The three components have demonstrated effectiveness in improving the overall selection performance rather than solely focusing on enhancing R@1.

As demonstrated in Table 10, our proposed node attention and the corresponding dot product calculation effectively enhance R@1/5. This suggests that the node representations obtained through the aggregation of knowledge embeddings possess robust expressive capabilities. The scoring mechanism during the agent's traversal process and the corresponding selection strategy are effective, particularly evident in the case of WoW Test Unseen, which suggests that the guiding role of node scores in knowledge selection remains effective even for zero-shot topics. The decrease in R@10 performance could be attributed to the amplification of knowledge score variance by node attention. Moreover, the enhancement in knowledge selection accuracy contributes to improving the quality of response generation.

## D    Case Study for WoW

To ensure the credibility of the comparisons, we conducted a case study using samples previously discussed in the relevant literature (Bai et al., 2023). As shown in Table 4, the dialogue topic is "Veterinary physician" and the Wizard is asked the question, "What makes you want to be a veterinarian?".

In the ground-truth data, the Wizard responds with "wanting to help animals" based on the description of the veterinary role in selected knowledge. In contrast, MIKE's generated response incorrectly assumes that the Wizard is already a veterinarian, disregarding the dialogue context. KnowledGPT's response lacks confidence and fails to provide adequate reasoning. While KINET offers reasons for pursuing a career as a veterinarian, its emphasis on the "clinical environment" does not align with the preceding context and relevant knowledge. Conversely, GATE provides a comprehensive and contextually appropriate answer regarding "becoming a veterinarian," demonstrating its capability to effectively select suitable knowledge for generating coherent and engaging responses.

| | WoW | | | | | | OpenDialKG | | | | | |
|---|---|---|---|---|---|---|---|---|---|---|---|---|
| | Train | Valid-S | Valid-U | Test-S | Test-U | Total | Train | Valid-S | Valid-U | Test-S | Test-U | Total |
| Conversations | 18,430 | 981 | 967 | 965 | 968 | 22,311 | 10,969 | 1,176 | 1,176 | 1,176 | 1,176 | 15,673 |
| Utterances | 166,787 | 8,921 | 8,794 | 8,715 | 8,782 | 201,999 | 63,832 | 6,845 | 6,839 | 6,842 | 6,851 | 91,209 |
| Avg. Knowledge | 75.48 | 75.27 | 76.90 | 75.24 | 74.59 | – | 600.0 | 590.2 | 600.7 | 603.5 | 610.4 | – |

Table 6: The statistics of WoW and OpenDialKG datasets.

| Prompt – Response Generation | |
|---|---|
| Assuming there is a seeker of knowledge who engages in a conversation (named "apprentice" / "user") with a wise person who has access to knowledge (named "wizard" / "assistant"), I will provide the history of their conversation and the available reference knowledge as follows: | |
| **History of conversation:** | [ History ] |
| **Reference knowledge:** | [ Knowledge ] |
| As the wizard/assistant, please continue the dialogue with the apprentice/user, keeping in mind the history of their conversation + ["K"] + Provide a response of less than 20 words. | ["K"](ChatGPT + GATE): and the available reference knowledge. ["K"](ChatGPT Only): and leveraging your knowledge. |

Table 7: Prompts used for response generation. Different ["K"] represent different response settings, as mentioned in Section 4.5.

| Prompt – Evaluation | |
|---|---|
| Please evaluate the engagement/informativeness level of the following responses and rank them. | |
| **History of conversation:** | [ History ] |
| **Response A:** | [ Response ] |
| **Response B:** | [ Response ] |
| Please provide a detailed explanation of your thought process, outlining each step you take to arrive at your conclusion, before providing your answer to the question. | |

Table 8: Prompts used for evaluation of engagement and informativeness.

| | WoW | | | | | | OpenDialKG | | | | | |
|---|---|---|---|---|---|---|---|---|---|---|---|---|
| | Test Seen | | | Test UnSeen | | | Test Seen | | | Test UnSeen | | |
| | R@1 | R@5 | R@10 | R@1 | R@5 | R@10 | R@1 | R@5 | R@10 | R@1 | R@5 | R@10 |
| **GATE** | **31.81** | **60.02** | **69.05** | **27.60** | 57.84 | **72.25** | **30.21** | 54.96 | 62.33 | **27.72** | **56.85** | **61.01** |
| w/o $\mathcal{L}_{Walk}$ | 30.42 | 56.65 | 67.32 | 26.19 | 58.19 | 70.79 | 29.26 | **56.16** | **62.80** | 26.90 | 53.27 | 59.51 |
| w/o $\mathcal{L}_{Node}$ | 31.23 | 57.05 | 67.11 | 27.09 | **59.45** | 71.49 | 26.68 | 49.59 | 53.54 | 24.53 | 46.66 | 51.43 |
| w/o $\mathcal{L}_{Knowledge}$ | 2.366 | 7.624 | 12.36 | 2.273 | 7.231 | 12.14 | 0.765 | 3.201 | 6.163 | 0.454 | 2.508 | 4.873 |
| w/ $\mathcal{L}_{Walk}$ | 0.894 | 5.573 | 13.91 | 0.491 | 5.062 | 15.11 | 0.573 | 3.798 | 6.976 | 0.717 | 3.153 | 6.307 |
| w/ $\mathcal{L}_{Node}$ | 2.471 | 7.282 | 12.64 | 2.247 | 7.180 | 12.78 | 0.526 | 3.345 | 6.761 | 0.669 | 2.437 | 5.471 |
| w/ $\mathcal{L}_{Knowledge}$ | 31.62 | 55.76 | 64.64 | 26.42 | 57.64 | 68.80 | 27.88 | 51.43 | 55.92 | 24.77 | 48.42 | 52.87 |

Table 9: The ablation results for the loss function.

| | WoW | | | | | | OpenDialKG | | | | | |
|---|---|---|---|---|---|---|---|---|---|---|---|---|
| | Test Seen | | | Test UnSeen | | | Test Seen | | | Test UnSeen | | |
| | Knowledge Selection | | | | | | | | | | | |
| | R@1 | R@5 | R@10 | R@1 | R@5 | R@10 | R@1 | R@5 | R@10 | R@1 | R@5 | R@10 |
| GATE | **31.81** | **60.02** | 69.05 | **27.60** | **57.84** | **72.24** | **30.21** | **54.96** | 62.33 | **27.72** | 56.85 | 61.01 |
| - Node Attention | 29.86 | 57.81 | **71.45** | 18.62 | 50.00 | 68.88 | 30.03 | 52.99 | **65.03** | 25.63 | **57.29** | **61.40** |
| | Response Generation | | | | | | | | | | | |
| | Rouge | Meteor | F1 | Rouge | Meteor | F1 | Rouge | Meteor | F1 | Rouge | Meteor | F1 |
| GATE | **24.20** | **23.12** | **26.23** | **23.81** | **22.78** | **25.75** | **28.57** | **28.46** | **32.21** | **27.93** | **27.78** | **31.57** |
| - Node Attention | 23.98 | 23.00 | 26.20 | 22.09 | 21.11 | 24.29 | 28.15 | 28.01 | 31.7 | 27.62 | 27.67 | 31.37 |

Table 10: The ablation results for node attention.

| History | Wizard: | I want to be a veterinary physician when I grow up. |
|---|---|---|
| | Apprentice: | That's really ambitious. I started school to be a vet, but I ended up being a teacher. What makes you want to be a veterinarian? |
| Ground-truth Response | Wizard: | I want to treat animal illness and perform surgery to help them. |
| Ground-truth Knowledge | "Veterinary physician": | In many cases, the activities that may be undertaken by a veterinarian (such as treatment of illness or surgery in animals) are restricted only to those professionals who are registered as a veterinarian. |
| MIKE: | I am a veterinarian. I know that most veterinary physicians work in clinical settings, treating animals directly. | |
| KnowleGPT: | I'm not sure how I'd like to be a veterinarian. I think I would like to treat animals directly. | |
| KINET: | I love animals and I want to work in clinical settings treating them directly. | |
| GATE | I want to treat diseases, disorders and injuries in animals. | |

Table 11: Case Study on WoW dataset, baselines' responses are reported from Bai et al. (2023).