# OpenReview forum: "Well Begun is Half Done: Generator-agnostic Knowledge Pre-Selection for Knowledge-Grounded Dialogue"
_EMNLP/2023/Conference — EMNLP 2023 Main_

### Official Review · Reviewer_AqSZ · 2023-07-27

**Soundness:** 4

**Excitement:**

4: Strong: This paper deepens the understanding of some phenomenon or lowers the barriers to an existing research direction.

**Paper Topic And Main Contributions:**

This paper proposes a generator-agnostic knowledge pre-selection method for knowledge-grounded dialogue. Their method firstly unify different knowledge structures, and then knowledge is selected and adapted to appropriate size based on GAT and reinforcement learning. The authors apply their method on both pre-trained models and large language models for knowledge-grounded dialogues and make significant improvement.

**Reasons To Accept:**

1. The proposed knowledge pre-selection method is an essential topic especially in the context of LLM. As language model can hardly encode all knowledge accurately in its storage, using a knowledge base to augment its generation is a progressing direction.
2. The taxonomy of different manners to perform knowledge augmentation is inspirational, and the experiment design and analysis is in-detail and well structured.
3. Both unification of different knowledge structures and the adaptation of knowledge pool size are interesting.

**Reasons To Reject:**

1. As LLM is taking the dialogue generation area by storm, the experiment should be more focused on LLM-based methods.

**Reproducibility:**

4: Could mostly reproduce the results, but there may be some variation because of sample variance or minor variations in their interpretation of the protocol or method.

**Reviewer Confidence:**

4: Quite sure. I tried to check the important points carefully. It's unlikely, though conceivable, that I missed something that should affect my ratings.

---

> ### Author Rebuttal · Authors · 2023-08-29
>
> Thank you for your positive comments.
>
> According to your suggestion *"experiment should be more focused on LLM-based methods"*, we believe that the CoT technology can be used to create prompts tailored to this scenario, enabling precise guidance for LLMs to utilize pre-selected knowledge for higher-quality responses. This enhances the effectiveness of the Knowledge Pre-Selection approach as an aid to LLMs. Furthermore, exploring the assessment of engagement and informativeness by leveraging LLMs is also a valuable avenue for future research. Your suggestions have greatly inspired our forthcoming research.

---

### Official Review · Reviewer_7Yzk · 2023-08-04

**Soundness:** 3

**Excitement:**

3: Ambivalent: It has merits (e.g., it reports state-of-the-art results, the idea is nice), but there are key weaknesses (e.g., it describes incremental work), and it can significantly benefit from another round of revision. However, I won't object to accepting it if my co-reviewers champion it.

**Paper Topic And Main Contributions:**

The paper proposes a generator-agnostic knowledge pre-selection method called GATE for knowledge-grounded dialogue systems. The paper focuses on the  knowledge selection before generation and demonstrates the superiority of GATE in selecting context-related knowledge and facilitating response generation models to generate more informative responses. The experimental results on two datasets show that GATE outperforms other baselines in terms of knowledge selection accuracy and response quality.


**Questions For The Authors:**

1. Can you provide more implementation details, such as the hyperparameters and training process, to facilitate reproducibility?

2. Why do the two datasets in Table 3 have inconsistent Engaging performance with the addition of GATE?

**Reasons To Accept:**

1. GATE is a generator-agnostic method that can select context-related knowledge and improve the quality of response generation.

2. The experimental results demonstrate the superiority of GATE in knowledge selection and its ability to facilitate response generation models.

**Reasons To Reject:**

1. The paper lacks implementation details, making it difficult to reproduce the study.

2. The paper would benefit from a more detailed discussion of the limitations and future directions of the proposed method.

3. The author emphasizes several times that the pre-selection has been largely ignored by previous work, but there is actually a lot of work focused on knowledge selection:

[1] Kim, Byeongchang ,  J. Ahn , and  G. Kim . "Sequential Latent Knowledge Selection for Knowledge-Grounded Dialogue." arXiv (2020).

[2] Yang, Chenxu, et al. "TAKE: topic-shift aware knowledge selection for dialogue generation." Proceedings of the 29th International Conference on Computational Linguistics. 2022.

**Reproducibility:**

4: Could mostly reproduce the results, but there may be some variation because of sample variance or minor variations in their interpretation of the protocol or method.

**Reviewer Confidence:**

4: Quite sure. I tried to check the important points carefully. It's unlikely, though conceivable, that I missed something that should affect my ratings.

---

> ### Author Rebuttal · Authors · 2023-08-29
>
> Thank you for your detailed review. We will address your concerns point by point.
>
>  **Q1:** *"... lacks implementation details ..."* [Reasons to reject 1]
>
>    **A1:**
>    The activation function used in the model is LeakyReLU with a negative slope of 0.21. The dropout layer probability applied across the model is uniformly set at 0.25.
>    GATE is trained on a single RTX A5000, employing a learning rate 1e-4.
>    We will add the above supplementary details in the revised paper.
>
>  **Q2:** *"... would benefit from a more detailed discussion of the limitations and future directions of the proposed
> method"* [Reasons to reject 2]
>
>    **A2:**
>    There is room for improvement in the reward setting of our proposed GATE.
>    For example, considering the agent's walk through the graph, the variation of distance between the agent and the correct process node/knowledge node can be incorporated into the existing reward setting.
>    In addition, we will further explore the role of GATE as an aid to more LLM-based models.
>    For example, constructing scenario-specialized prompts and coupled with the CoT technique can guide LLMs more meticulously in utilizing the highly relevant explicit knowledge provided by GATE to generate higher-quality responses.
>
>  **Q3:** *"but there is actually a lot of work focused on knowledge selection"* [Reasons to reject 3]
>
>    **A3:**
>    We have not overlooked the category of knowledge pre-selection and have mentioned it several times in our paper (cf. Lines 168-184 and 417-423).
>    We thank the reviewer for the two relevant works and we will include them in our revised paper.
>    It needs to be clarified that the SKT model in [1] incorporates posterior information from responses during training, thus belongs to the "knowledge selection" category rather than the "pre-selection" category.
>
>  **Q4:** *"... provide more implementation details ..."* [Questions for the authors 1]
>
>    **A4:**
>    Please refer to A1 for more information.
>
>  **Q5:** *"inconsistent Engaging performance"* [Questions for the authors 2]
>
>    **A5:**
>    This inconsistency stems from varying knowledge quantities in datasets. The WoW dataset contains complete sentences, and 5/10 knowledge pieces are overly detailed, causing rigid responses and reduced engagement. Conversely, the OpenDialKG dataset has concise triplets. Appropriate information from 5/10 knowledge pieces leads to captivating responses and heightened engagement.

---

### Official Review · Reviewer_CwT6 · 2023-08-04

**Soundness:** 3

**Excitement:**

3: Ambivalent: It has merits (e.g., it reports state-of-the-art results, the idea is nice), but there are key weaknesses (e.g., it describes incremental work), and it can significantly benefit from another round of revision. However, I won't object to accepting it if my co-reviewers champion it.

**Paper Topic And Main Contributions:**

The paper focuses on the topic of knowledge selection. The authors compare various types of knowledge selection methods (pre-response generation, during response generation, post response generation). They find selecting knowledge before response generation performs the best.

The main contribution as I see it is developing a knowledge selection method that can access different types of knowledge bases (structured / unstructured).

**Questions For The Authors:**

In Table 4, the keyword 2008 is not in the knowledge sentence which would fall under the category of hallucination. How does your method compare against others in terms of this factor.

**Reasons To Accept:**

(1) The knowledge selection method that is supposed to handle both structured / unstructured knowledge is a good proposal and it seems results are sufficient to back that up.

(2) Nice to see results using ChatGPT and that knowledge selection can further improve it's performance and that it is still critical even for non-current events.

**Reasons To Reject:**

(1) The claim that there has been no work on selecting knowledge before response generation is not correct. Some relevant works are.

https://arxiv.org/pdf/2203.00763.pdf
https://arxiv.org/pdf/2104.07567.pdf

With that being said I do think that the method is able to retrieve knowledge in both a structured/unstructured setting is good.

(2) There are a few items in the paper that need to be clarified such as:

How are the keywords extracted from the conversation history?
The reward signals R_node and R_gold seem the same and it is not clear what the difference is between them.
Your method can choose variable number of knowledge sentences but Wizard of Wikipedia only has one gold knowledge sentence so how did you train it?
Explain the evaluation process on line 481 in more detail.

(3) It would be good to discuss why is pre-selection better than post selection. In post selection you already have the response so ideally it should be more accurate at selecting the knowledge.

**Reproducibility:**

4: Could mostly reproduce the results, but there may be some variation because of sample variance or minor variations in their interpretation of the protocol or method.

**Reviewer Confidence:**

4: Quite sure. I tried to check the important points carefully. It's unlikely, though conceivable, that I missed something that should affect my ratings.

**Typos Grammar Style And Presentation Improvements:**

In Table 4 "Paper towns" is highlighted red for the GATE response indicating it is incorrect but it should be highlighted green I believe.

---

> ### Author Rebuttal · Authors · 2023-08-29
>
> Thank you for your detailed review. We will address your concerns point by point.
>
> **Q1:** *"... there has been no work on selecting knowledge before response generation is not correct ..."*
>
> **A1:**
> There are indeed pieces of work that select for knowledge before response generation.
> We have not overlooked this type of work and have mentioned it several times in our paper (cf. Lines 168-184 and 417-423).
> We thank the reviewer for the two relevant works and we will include them in our revised paper.
>
> In addition, we need to restate the core contributions of our work, i.e., **the introduction of a novel perspective to organize the literature of knowledge selection in knowledge-grounded dialogue**, instead of proposing a pre-selection method for the first time (cf. Lines 003-006, 037-039, and 107-110).
> We hope that this perspective can offer valuable insights to this community.
>
>  **Q2:** *"There are a few items in the paper that need to be clarified"*
>
> **A2:**
> We provide clarifications for them as follows:
>
> a. *"How are the keywords extracted from the conversation history."*
>
> We leverage the KeyBert toolkit to extract up to five keywords from the conversation history by calculating the cosine similarity between document-level representation and word embeddings.
>
> b. *"The reward signals $R_{Node}$ and $R_{Gold}$ ... the difference ..."*
>
> $R_{Node}$ refers to the reward for the Agent when halting at the correct process node,
> while $R_{Gold}$ indicates the reward for GATE when eventually choosing the gold knowledge (i.e., knowledge node).
>
> c. *"... variable number of knowledge sentences but Wizard of Wikipedia only has one gold knowledge sentence so how did you train it."*
>
> In accordance with the objective of knowledge pre-selection, GATE is trained to create suitably sized knowledge pools rather than learning to select single Gold knowledge.
>     Given the assumption: "If the model selects the correct knowledge with a high probability, excessive knowledge can be deemed redundant. Otherwise, expanding the knowledge pool is necessary to enhance knowledge effectiveness'',
>     the pool size is determined by the variance of knowledge-scoring results, indicated by Equation (2): larger variance results in a smaller knowledge pool, and vice versa (cf. Line 279).
>
> d. *"... the evaluation process on line 481 ..."*
>
> We provide the conversation history to ChatGPT, present different responses (ChatGPT/ChatGPT+GATE), and randomly order them in the prompt. ChatGPT compares responses based on engagement and informativeness, offering justifications and a comparative analysis.
> Due to space constraints, the complete prompts are included in Appendix B.
>
>  **Q3:** *"... discuss why is pre-selection better than post selection ..."*
>
> **A3:**
> Thank you for pointing out this worthwhile discussion.
>
> Post-selection is dedicated to correcting potential knowledge errors in the generated response, and pre-selection performs knowledge selection before generation to potentially reduce the learning, adjustment, and interpretation burden of subsequent response generation models.
> These two categories of knowledge selection (pre- and post-) share the common pursuit of improving the quality of generated response, but they are executed in different ways at different stages of the process.
> Therefore, the two categories are not mutually exclusive and can be executed as a pipeline during the response generation process.
>
>  **Q4:** *"... the response ... should be more accurate at selecting the knowledge"*
>
> **A4:**
>
> In the post-selection category, the generated response can help in selecting knowledge to some extent.
> However, since this category is in an entity-replacement manner, there may be a mismatch between the selected knowledge and the generated response, leading to semantic and syntactic confusion.
>
> For example: an adjusted response in Table 4 with confusion --"It was released in __Meg Crosbie__",  where the bold entity "Meg Crosbie" is a mistake adjustment because a personal name does not make sense in the context of "released in".
>
> **Q5:** *"... Table 4, the keyword 2008 is ... hallucination. How does your method compare against others in terms of this factor."*
>
>    **A5:**
>    Thanks for your careful review.
>    We regret that the knowledge details selected by GATE and the two baselines are omitted in Table 4.
>    Please check the completed Table 4 below. K denotes selected knowledge, and R denotes the generated response.
>    We need to clarify that the "2008" keyword appearing in the response is not a hallucination, as it was selected from the knowledge base and does not contradict existing knowledge or reality.
>    Hallucination is a highly worthwhile topic but falls under the study of the GATE downstream task (i.e., response generation).
>
> | Dialogue Context       | (User) I heard about a movie called The Fault in Our Stars. Who stars in it?   |
> |------------------------|----------------------------------------------------------|
> |                        | (Agent) It's a good one! That was written by John Green and stars Shailene Woodley. |
> |                        | (User) Oh, I didn't know that. What else has John Green Written?                     |
> | **Ground-truth Response**  | He also wrote Paper Towns.                                 |
> | **Gold Knowledge**         | Paper Towns is written by John Green.                      |
> | **DiffKG**       | **K:** (A Confederacy of Dunces; written_by; Walker Percy)   |
> |                        | **R:** Sure! A Confederacy of Dunces is written by John Green. Do you like his work?|
> | **NPH**      | **K:** (Meg Crosbie; ~starred_actors; Paper Towns)   |
> |                        | **R:** Paper Towns is another one he wrote. It was released in Meg Crosbie.|
> | **GATE**       | **K:** (Paper Towns; written_by; John Green), (Paper Towns; release_year; 2008), ...   |
> |                        | **R:** Paper Towns is another one he wrote. It was released in 2008.|
>
>  **Q6:** *"``Paper towns'' ... should be highlighted green"*
>
>    **A6:**
>    We will modify this typo in our revised manuscript.

---

### Meta-Review · Area_Chair_mq1r · 2023-10-04

**Recommendation:** 4

**Metareview:**

A pipeline for the topic of knowledge selection. However, some reviewers think the author still has to address some of the points in the final version.

---

### Decision · Program_Chairs · 2023-10-07

**Decision:**

Accept-Main

**Comment:**

A pipeline for the topic of knowledge selection. However, some reviewers think the author still has to address some of the points in the final version.